# COVID-19 and crime: Analysis of crime dynamics amidst social distancing protocols

**Shelby M. Scott** [1]\*, **Louis J. Gross**[1,2]

**1** Department of Ecology and Evolutionary Biology, University of Tennessee, Knoxville, Tennessee, United States of America, **2** National Institute for Mathematical and Biological Synthesis, Knoxville, Tennessee, United States of America

\* sscott41@vols.utk.edu

## Abstract

In response to the pandemic in early 2020, cities implemented states of emergency and stay at home orders to reduce virus spread. Changes in social dynamics due to local restrictions impacted human behavior and led to a shift in crime dynamics. We analyze shifts in crime types by comparing crimes before the implementation of stay at home orders and the time period shortly after these orders were put in place across three cities. We find consistent changes across Chicago, Baltimore, and Baton Rouge with significant declines in total crimes during the time period immediately following stay at home orders. The starkest differences occurred in Chicago, but in all three cities the crime types contributing to these declines were related to property crime and statutory crime rather than interpersonal crimes.

**Data Availability Statement:** All relevant data are within the paper and its Supporting information files. Specific code and information files are available at https://github.com/shelbymscott/COVIDandCrime.

## Introduction

Crime is a major public concern in the United States. Gun violence leads to the deaths of 36,000 individuals and the non-fatal injuries of 85,000 others [1]. Other crime types can produce property damage, trauma for victims, and fractures in community trust [2]. Across the five major crime categories (personal, property, inchoate, statutory and financial), some rely on regular interpersonal interactions and social dynamics [3].

In early 2020, COVID-19 emerged in the United States and case numbers quickly grew across the country. In response to the pandemic, many cities implemented a variety of mitigation policies to minimize spread. A major strategy for preventing viral spread focuses on reducing contacts between individuals using stay at home orders, which closed non-essential businesses, mandated wearing masks, and encouraged citizens to minimize all non-essential visits [4]. Adherence to these policies was initially strong, but faded over time, as evidenced by self-reporting and cell phone movement data (see S1-S3 Figs in S1 File) [5]. Implementing these types of policies resulted in drastic changes in the behaviors of citizens. Because crime is associated with human behavior, many have questioned how reductions in human contact have changed the dynamics of crimes across the country [6–15].

In a pandemic which strains health care resources, collateral mortality may increase, due to the inability of the health care system to respond as effectively to crime-derived injuries [16].

**Funding:** LJG is supported by the National Institute for Mathematical and Biological Synthesis (NIMBioS), which is funded through NSF Award #DBI-1300426, with additional support from The University of Tennessee, Knoxville (http://www.nimbios.org/). SMS is funded through the National Defense Science and Engineering Graduate (NDSEG) Fellowship (https://ndseg.sysplus.com/).

**Competing interests:** The authors have declared that no competing interests exist.

Evaluating the impact of the pandemic on crime in a rigorous manner can assess the potential for significant changes in crime-derived mortality.

We test whether there are significant differences in the crime dynamics in Chicago, IL when comparing the time periods pre- and post-establishment of the stay at home orders, and then compare these results to Baton Rouge, LA and Baltimore, MD. We choose to analyze the first two weeks following the stay ay home order implementation because adherence to the mandates was strongest immediately following implementation (see S1-S3 Figs in S1 File) [5]. Our main focus is Chicago, due to its high rates of both COVID-19 and crime. Comparison of the crime dynamics in the three cities determines whether similar patterns exist (Fig 1). We hypothesize that changes in crime dynamics in all three cities following the implementation of stay at home orders are not uniform across all crime types and that there are differences between crimes associated with property, those which are statutory, and those involving inter-personal interactions. We define property crimes (denoted with P for this study) as those which involve interference with the property of another [3]. We define statutory crimes (denoted with S for this study) as those which are proscribed by statute [3]. We define inter-personal crimes (denoted with I for this study) as those that result in physical or mental harm to another person [3]. We find that total crimes declined significantly in the two weeks following implementation of stay-at-home orders across all three cities. In Chicago total crimes declined 31.5%. The property crimes with significant declines were burglary (22.9%), criminal trespass (50.1%), robbery (25.8%), and theft (41.0%). The statutory crimes with significant declines were interference with public officers (93.1%), narcotics (86.1%), and other offenses (41.2%). The interpersonal crimes with significant declines were assault (19.4%), and criminal sexual assault (56.0%). Baltimore also experienced significant declines in total crimes (25.9%). The crime types showing significant declines were auto theft (29.9%), burglary (29.2%), and larceny (35.0%), which are all property crimes. Baton Rouge also experienced significant declines in total crimes (22.4%). The crime types that showed significant declines were narcotics (52.3%) and other crimes (41.5%), which are both statutory crimes.

## Materials and methods

### Data

Chicago, Baltimore, and Baton Rouge each have publicly available crime datasets through city-wide data portals [17–19]. Due to lack of consensus on collecting, defining, and reporting crime, care should be taken when comparing one dataset to another. Descriptions of each crime type included are available in the (see S2 Table in S1 File). These three cities were chosen because they had openly available crime data and they differ in demographics (see S1 Table in S1 File) [20]. Chicago has a 4.5 times larger population than Baltimore and 12 times larger than Baton Rouge [20]. All three cities have fairly high poverty rates and regularly receive media attention for their crime dynamics. Crime in all three cities remained fairly consistent from 2017-2019 (see S4-S6 Figs in S1 File). They also exist in three different regions of the country, making them interesting comparisons for this study [20].

All data and code files used in this analysis are available through GitHub at https://github.com/shelbymscott/COVIDandCrime.

**Chicago data.** The Chicago City Data Portal provides crime reports from 2001-present [19]. Data are extracted from the CLEAR (Citizen Law Enforcement Analysis and Reporting) system. The information from this dataset used for this analysis includes the date, primary type, and description, but a number of other items are available. We isolated the crimes which occurred from January 1, 2020 through April 4, 2020, which gave data from before the onset of the pandemic (01/01/2020—03/08/2020), during the state of emergency (03/09/2020—03/20/

(a)

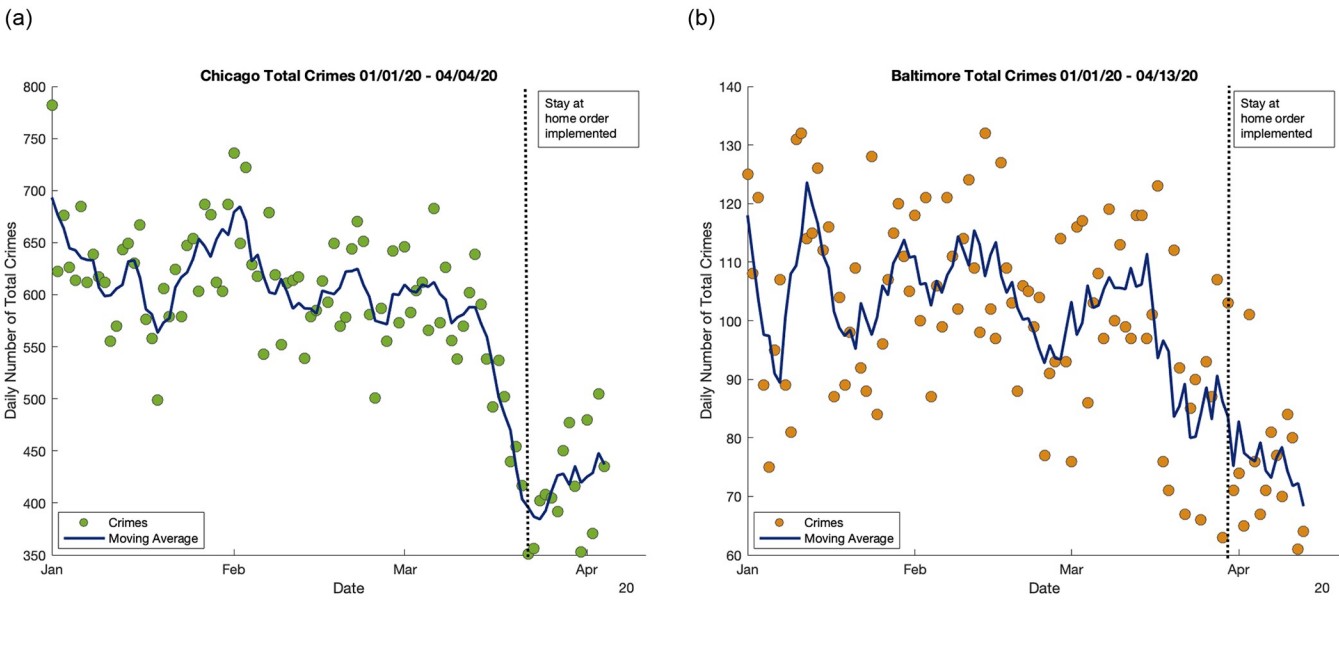

(c)

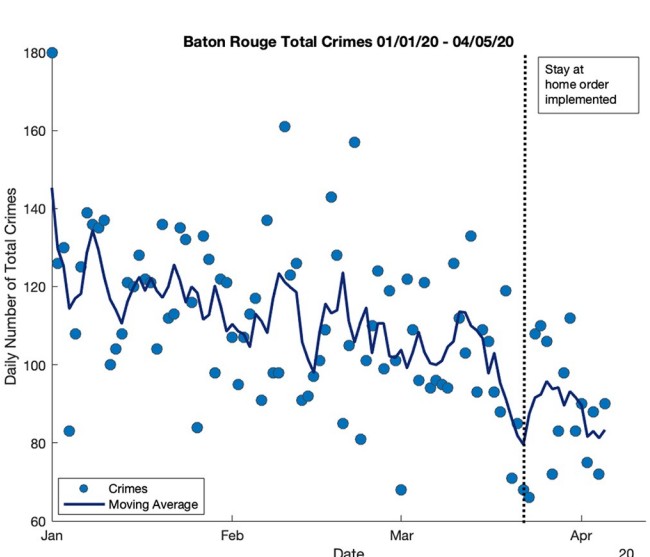

**Fig 1. The total crime trends seen in Chicago, Baltimore, and Baton Rouge from the beginning of the year through the end of our study period.** Each point is the total number of crimes observed on that day. The vertical line represents the day when the stay at home order was implemented for each city. The dark line represents a moving average of the data with $k = 5$ to observe shifts in dynamics over a five day temporal window [23]. More information regarding these methods is available in the S1 File.

2020), and after the stay at home order was put in place (03/21/2020—04/04/2020) for temporal comparisons. There are 32 different crime types available, but for this analysis we choose 19 of the crime types with high occurrence and denote them as property (P), statutory (S), or interpersonal (I). They include:

- Total Crimes

- Gun Crimes (pulled from crime descriptions that included firearm use)

- Arson (P)

- Burglary (P)

- Criminal Damage (P)

- Criminal Trespass (P)

- Robbery (P)

- Theft (P)

- Weapons Violation (P)

- Interference with Public Officer (S)

- Narcotics (S)

- Other Offense (S)

- Public Peace Violation (S)

- Assault (I)

- Battery (I)

- Criminal Sexual Assault (I)

- Homicide (I)

- Sex Offense (I)

These crime types were chosen because they composed a high proportion of total crimes or were present in the Baltimore or Baton Rouge datasets for potential comparison. Definitions of these crime types are available in the (see S2 Table in S1 File). Those crime types excluded due to low numbers were: concealed carry license violations, deceptive practices, gambling, human trafficking, intimidation, kidnapping, liquor law violations, obscenity, offenses involving children, other narcotics violations, prostitution, public indecency, and stalking. We determined whether the crime types should be included based on if they comprised greater than or equal to 0.2% of total crimes. This number was found by determining the lowest percentage from the Baltimore and Baton Rouge datasets and only including those crime types which exceeded this amount.

**Baltimore data.** Baltimore, Maryland has Victim Based Crime data available for public download [18]. The data are preliminary and therefore may be subject to change [18]. The data provided includes a number of different options from which to choose. The relevant information we use includes the date of the crime and the crime type. We isolate those crime types which occurred from the beginning of the year (1/1/20) to when the stay at home order was put in place (3/29/20) and the two weeks following the implementation (3/30/20—4/13/20). The 11 crime types available for analysis are denoted as property (P) or interpersonal (I) as this dataset did not account for statutory crimes since it only provides crimes where an individual was victimized. Definitions of these crime types are available in the (see S2 Table in S1 File). The dataset includes:

- Total Crimes

- Gun Crimes (determined by finding those crimes which involved a firearm as a weapon)

- Arson (P)

- Auto Theft (P)

- Burglary (P)

- Larceny (P)

- Robbery (P)

- Assault (I)

- Homicide (I)

- Rape (I)

- Shooting (I)

**Baton Rouge data.** Crimes reported in Baton Rouge are handled by the Baton Rouge Police Department and are publicly available [17]. The data are pulled from police reports using automatic statistical reporting, which can lead to some errors [17]. We obtained the dates and crime types from the dataset and parse for the time period from the beginning of the year to when the stay at home order was put in place (1/1/20—3/21/20) and the two weeks following the stay at home order (3/22/20—4/5/20). The 15 crime types analyzed are denoted as property (P), statutory (S), and interpersonal (I). Definitions of these crime types are available in the (see S2 Table in S1 File). The dataset includes:

- Total Crimes

- Burglary (P)

- Criminal Damage (P)

- Robbery (P)

- Theft (P)

- Juvenile (S)

- Narcotics (S)

- Nuisance (S)

- Other (S)

- Vice (S)

- Assault (I)

- Battery (I)

- Firearm (I)

- Homicide (I)

- Sexual Assault (I).

### *t*-test analysis

In order to determine whether the differences seen in crime dynamics are significant or not, we performed numerous *t*-tests that compared whether different years and time periods

differed significantly [21]. *T*-tests determine whether the means of two sets of data are significantly different from each other [21]. Because we were performing a number of statistical tests simultaneously, we used the Bonferroni correction to be sure that spurious positives were not included by random chance [22]. The Bonferroni correction sets the significance value, $\alpha$, for the entire set of $n$ *t*-tests run equal to $\alpha^*$ by taking:

$$\alpha* = \frac{\alpha}{n}.$$

Formally, given $n$ *t*-tests $T_i$ for hypotheses $H_i$ (*i* between 1 and $n$) under the assumption $H_0$ that all hypotheses $H_i$ are false, and if the individual test critical values are less than or equal to $\alpha/n$, then the critical value is less than or equal to $\alpha$. In equation form, if:

$$P(T_i \text{ passes } |H_0) \leq \frac{\alpha}{n}$$

for $1 \leq i \leq n$, then

$$P(\text{some } T_i \text{ passes}|H_0) \leq \alpha,$$

which follows from the Bonferroni inequalities [21]. The *p*-value result from a *t*-test represents the probability of obtaining test results at least as extreme as the results actually observed, under the assumption that the null hypothesis is correct [21].

There were *t*-tests performed on the datasets in order to determine whether what has been observed in 2020 is significantly different behavior from other years. These included:

- Year to year tests to determine whether the 2020 observed behavior in Chicago differs from what we would have expected from previous years (Table 1, S4-S6 Figs in S1 File).

- Within year analysis to determine whether the behavior before pandemic response (1/1/20—3/20/20) and after the stay at home orders were put in place in Chicago (3/21/20—4/3/20) differ from one another (Table 2, S7-S10 Figs in S1 File).

- Time period comparisons over Baltimore and Baton Rouge to determine whether the patterns seen in response to COVID in Chicago are consistent across numerous cities (Table 2, S11 and S12 Figs in S1 File).

- Comparison tests with past years to determine whether the observed behavior is due to seasonality or differing temporal dynamics during the pandemic (S3-S19 Tables in S1 File, S4-S6 Figs in S1 File).

## Results

Comparing January through early April Chicago crime data (early 2020) to the same time period in each of the three previous years determines which crime types in 2020 are significantly different [19]. The results (Table 1, S3-S5 Tables and S9-S17 Tables in S1 File) show that between the early months of 2019 and 2020, there were decreases in total crimes, burglaries, narcotics, other offenses, and thefts. Between 2018 and 2020, total crimes, burglaries, criminal damages, criminal trespasses, narcotics, other offenses, robberies, and thefts all decreased, while weapons violations increased. Between 2017 and 2020, total crimes, burglaries, criminal damages, criminal trespasses, other offenses, public peace violations, robberies, and thefts all decreased, while weapons violations increased. These comparisons include the time period from January 1 to April 4.

**Table 1. Comparisons of each Chicago crime type in the first three months 2019, 2018, and 2017 compared to crime types in the same time period of 2020.** The degrees of freedom for these analyses are 179 and $\alpha = .05$ is adjusted after Bonferroni correction (with $n = 18$) to $\alpha = 0.0027$. The values for mean $\mu$, standard deviation $\sigma$, and percent change are also provided. Bolded crime types show significant differences between years and crime categories are denoted by (P) for property, (S) for statutory, and (I) for interpersonal crimes.

| | Early 2019 Dataset ($\mu$, $\sigma$) | Early 2020 Dataset ($\mu$, $\sigma$) | p-value | Percent Change |
|---|---|---|---|---|
| | **Total Crimes (650, 84.3)** | **Total Crimes (575, 90.6)** | **$7.45 \times 10^{-7}$** | **-11.5%** |
| | Gun Crimes (33.6, 9.37) | Gun Crimes (36.4, 9.38) | 0.038 | 8.33% |
| P | Arson (0.872, 0.845) | Arson (0.884, 1.07) | 0.933 | -1.38% |
| P | **Burglary (23.7, 7.37)** | **Burglary (20.6, 5.16)** | **$8.53 \times 10^{-4}$** | **-13.1%** |
| P | Criminal Damage (61.7, 13.8) | Criminal Damage (59.1, 12.4) | 0.167 | -4.40% |
| P | Criminal Trespass (17.5, 5.02) | Criminal Trespass (16.1, 5.70) | 0.076 | -8.00% |
| P | Robbery (18.4, 5.59) | Robbery (20.4, 6.11) | 0.019 | 10.9% |
| P | **Theft (149, 23.7)** | **Theft (130, 28.9)** | **$1.09 \times 10^{-6}$** | **-12.8%** |
| P | Weapons Violation (14.5, 5.93) | Weapons Violation (15.6, 5.80) | 0.186 | 7.59% |
| S | Interference with Public Officer (3.40, 1.65) | Interference with Public Officer (3.32, 2.19) | 0.755 | -2.35% |
| S | **Narcotics (44.1, 11.0)** | **Narcotics (31.0, 13.6)** | **$8.41 \times 10^{-12}$** | **-29.7%** |
| S | **Other Offense (47.8, 9.80)** | **Other Offense (39.4, 9.98)** | **$2.89 \times 10^{-8}$** | **-17.6%** |
| S | Public Peace Violation (3.81, 2.18) | Public Peace Violation (2.87, 2.06) | 0.003 | -24.7% |
| I | Assault (50.8, 9.89) | Assault (47.4, 8.02) | 0.010 | -6.69% |
| I | Battery (122, 22.5) | Battery (115, 21.7) | 0.033 | -5.74% |
| I | Criminal Sexual Assault (3.81, 2.44) | Criminal Sexual Assault (3.73, 2.47) | 0.818 | -2.10% |
| I | Homicide (0.989, 1.05) | Homicide (1.08, 1.03) | 0.534 | 9.2% |
| I | Sex Offense (3.28, 2.37) | Sex Offense (2.78, 1.65) | 0.095 | -15.2% |
| | Early 2018 Dataset | Early 2020 Dataset | | |
| | **Total Crimes (655, 65.2)** | **Total Crimes (575, 90.6)** | **$7.63 \times 10^{-11}$** | **-12.2%** |
| | Gun Crimes (35.1, 7.99) | Gun Crimes (36.4, 9.38) | 0.301 | 3.70% |
| P | Arson (0.787, 0.914) | Arson (0.884, 1.07) | 0.504 | 12.3% |
| P | **Burglary (27.6, 6.31)** | **Burglary (20.6, 5.16)** | **$1.14 \times 10^{-14}$** | **-25.4%** |
| P | **Criminal Damage (66.2, 14.8)** | **Criminal Damage (59.1, 12.4)** | **$4.00 \times 10^{-4}$** | **-10.7%** |
| P | **Criminal Trespass (19.0, 3.78)** | **Criminal Trespass (16.1, 5.70)** | **$5.32 \times 10^{-5}$** | **-15.3%** |
| P | **Robbery (25.1, 6.38)** | **Robbery (20.4, 6.11)** | **$7.45 \times 10^{-7}$** | **-18.7%** |
| P | **Theft (150, 21.1)** | **Theft (130, 28.9)** | **$1.19 \times 10^{-7}$** | **-13.3%** |
| P | **Weapons Violation (12.1, 4.37)** | **Weapons Violation (15.6, 5.80)** | **$7.84 \times 10^{-6}$** | **28.9%** |
| S | Interference with Public Officer (3.30, 1.75) | Interference with Public Officer (3.32, 2.19) | 0.951 | 0.606% |
| S | **Narcotics (37.3, 9.65)** | **Narcotics (31.0, 13.6)** | **$3.66 \times 10^{-4}$** | **-16.9%** |
| S | **Other Offense (45.4, 7.94)** | **Other Offense (39.4, 9.98)** | **$1.11 \times 10^{-5}$** | -13.2% |
| S | Public Peace Violation (3.22, 1.79) | Public Peace Violation (2.87, 2.06) | 0.214 | -10.9% |
| I | Assault (48.7, 9.40) | Assault (47.4, 8.02) | 0.295 | -2.67% |
| I | Battery (119, 19.9) | Battery (115, 21.7) | 0.137 | -3.36% |
| I | Criminal Sexual Assault (3.74, 2.67) | Criminal Sexual Assault (3.73, 2.47) | 0.961 | -0.267% |
| I | Homicide (1.30, 1.18) | Homicide (1.08, 1.05) | 0.188 | -16.9% |
| I | Sex Offense (2.61, 3.23) | Sex Offense (2.78, 1.65) | 0.644 | 6.51% |
| | Early 2017 Dataset | Early 2020 Dataset | | |
| | **Total Crimes (686, 74.7)** | **Total Crimes (575, 90.6)** | **$7.48 \times 10^{-17}$** | **-16.2%** |
| | Gun Crimes (37.7, 8.86) | Gun Crimes (36.4, 9.38) | 0.336 | 3.45 |
| P | Arson (1.21, 1.15) | Arson (0.884, 1.07) | 0.044 | -26.9% |
| P | **Burglary (35.5, 10.6)** | **Burglary (20.6, 5.16)** | **$6.24 \times 10^{-26}$** | **-42.0%** |
| P | **Criminal Damage (74.8, 14.1)** | **Criminal Damage (59.1, 12.4)** | **$6.27 \times 10^{-14}$** | **-21.0%** |
| P | **Criminal Trespass (18.5, 4.54)** | **Criminal Trespass (16.1, 5.70)** | **0.002** | **-13.0%** |

*(Continued)*

**Table 1.** (Continued)

| | | | | |
|---|---|---|---|---|
| P | **Robbery (29.5, 7.27)** | **Robbery (20.4, 6.11)** | $4.86 \times 10^{-17}$ | **-30.8%** |
| P | **Theft (154, 21.4)** | **Theft (130, 28.9)** | $5.99 \times 10^{-10}$ | **-15.6%** |
| P | **Weapons Violation (10.8, 4.23)** | **Weapons Violation (15.6, 5.80)** | $7.26 \times 10^{-10}$ | **44.4%** |
| S | Interference with Public Officer (2.77, 1.47) | Interference with Public Officer (3.32, 2.19) | 0.044 | 19.9% |
| S | Narcotics (33.9, 8.77) | Narcotics (31.0, 13.6) | 0.091 | -8.55% |
| S | **Other Offense (49.9, 8.96)** | **Other Offense (39.4, 9.98)** | $1.41 \times 10^{-12}$ | **-21.0%** |
| S | **Public Peace Violation (3.87, 2.12)** | **Public Peace Violation (2.87, 2.06)** | **0.001** | **-25.8%** |
| I | Assault (46.4, 8.24) | Assault (47.4, 8.02) | 0.386 | 2.16 |
| I | Battery (121, 19.6) | Battery (115, 21.7) | 0.037 | -4.96% |
| I | Criminal Sexual Assault (4.07, 3.56) | Criminal Sexual Assault (3.73, 2.47) | 0.436 | -7.62% |
| I | Homicide (1.65, 1.66) | Homicide (1.08, 1.04) | 0.006 | -34.5% |
| I | Sex Offense (2.67, 3.78) | Sex Offense (2.78, 1.65) | 0.798 | 4.12% |

To determine whether there were significant differences in crime between time periods, we performed a series of *t*-tests [21, 22]. While Chicago had three distinct time periods: pre-COVID, a state of emergency, and a stay at home order, the results from the state of emergency were not as stark as comparisons between the time period before the stay at home order was implemented and the two weeks after implementation (see supplementary text, S3-S17 Tables in S1 File, and S4 Fig in S1 File). Total crimes, assaults, burglaries, criminal sexual assaults, criminal trespasses, interference with public officers, narcotics, other offenses, robberies, and thefts each had significant decreases from the time period before the stay at home order was put in place to the two weeks following implementation (Table 2, Fig 2a).

In order to determine whether similar patterns have been observed in other cities with different demographics (S1 Table in S1 File), we test victim-based crime data from Baltimore, before the stay at home order was put in place and two weeks after implementation (Table 2) [18]. We find that total crimes, auto thefts, burglaries, and larceny all showed significant declines between the two time periods (Fig 2b).

As a further comparison to a smaller population city with different demographics (S1 Table in S1 File), crime data from Baton Rouge were analyzed (Table 2) [17]. We find that total crimes, narcotics, and other crimes all decreased significantly after the stay at home order was put in place (Fig 2c).

## Discussion and conclusions

The analysis of crime data from three cities indicates significant impacts on certain crime types arising from changes in social dynamics due to regulations implemented in response to the COVID-19 pandemic. The results of pair-wise *t*-tests, appropriately corrected, show that the crime dynamics experienced in 2020 significantly differ from previous years and that the implementation of strict stay at home orders in all three cities initially impacted crime.

Chicago's total crimes during the first three months of the year (including the time period following the stay at home order) declined in 2020 compared to 2017, 2018, and 2019 (see S3-S17 Tables in S1 File). Before the time period during which COVID protocols were put in place, there were no significant changes in total crimes compared to 2018 and 2019. There were some changes in particular crime types that may have been due to social factors, policing protocols, or other mechanisms occurring in past years. There were significant changes in total crimes between 2017 and 2020, with all significant crime types but weapons violations showing declines (S11 Table in S1 File). Once the stay at home order was put in place, there

**Table 2. Comparisons of the time period before the stay at home order was put in place and the two weeks after it was put in place in Chicago, Baltimore, and Baton Rouge.** The observed time period for the stay at home order spans from 03/21/2020—04/04/2020 in Chicago. The degrees of freedom for these analyses are 89 and $\alpha = .05$ is adjusted after Bonferroni correction (with $n = 18$) to $\alpha = 0.0027$. The two weeks after Baltimore's stay at home period span from 03/30/20—04/13/20. The degrees of freedom for these analyses are 101 and $\alpha = .05$ is adjusted after Bonferroni correction (with $n = 11$) to $\alpha = 0.0045$. The two weeks after Baton Rouge's stay at home order span from 03/22/20—04/05/20. The degrees of freedom for these analyses are 93 and $\alpha = .05$ is adjusted after Bonferroni correction (with $n = 15$) to $\alpha = 0.0033$. The values for mean $\mu$, standard deviation $\sigma$, and percent change are also provided. Bolded crime types show significant differences between years and crime categories are denoted by (P) for property, (S) for statutory, and (I) for interpersonal crimes.

| | Crime Types Pre-Stay at Home Orders $(\mu, \sigma)$ | Crime Types Post Stay at Home Orders $(\mu, \sigma)$ | p-value | Percent Change |
|---|---|---|---|---|
| | **Chicago** | | | |
| | **Total Crimes (606, 59.4)** | **Total Crimes (415, 47.8)** | $\mathbf{4.61 \times 10^{-20}}$ | **-31.5%** |
| | Gun Crimes (37.1, 9.47) | Gun Crimes (33.1, 8.42) | 0.139 | -10.8% |
| P | Arson (0.863, 1.09) | Arson (1.00, 1.00) | 0.651 | 15.9% |
| P | **Burglary (21.4, 5.08)** | **Burglary (16.5, 3.52)** | $\mathbf{6.89 \times 10^{-4}}$ | **-22.9%** |
| P | Criminal Damage (59.6, 12.5) | Criminal Damage (56.5, 11.5) | 0.375 | -5.20% |
| P | **Criminal Trespass (17.5, 5.08)** | **Criminal Trespass (8.73, 2.05)** | $\mathbf{3.38 \times 10^{-9}}$ | **-50.1%** |
| P | **Robbery (21.3, 5.88)** | **Robbery (15.8, 5.28)** | **0.001** | **-25.8%** |
| P | **Theft (139, 21.3)** | **Theft (82.0, 11.0)** | $\mathbf{1.83 \times 10^{-16}}$ | **-41.0%** |
| P | Weapons Violation (15.8, 6.02) | Weapons Violation (14.3, 4.42) | 0.339 | -9.49% |
| S | **Interference with Public Officer (3.89, 1.89)** | **Interference with PublicOfficer (0.267, 0.458)** | $\mathbf{7.46 \times 10^{-11}}$ | **-93.1%** |
| S | **Narcotics (35.9, 8.07)** | **Narcotics (5.00, 2.56)** | $\mathbf{7.09 \times 10^{-26}}$ | **-86.1%** |
| S | **Other Offense (42.2, 8.15)** | **Other Offense (24.8, 4.54)** | $\mathbf{3.27 \times 10^{-12}}$ | **-41.2%** |
| S | Public Peace Violation (3.14, 2.11) | Public Peace Violation (1.47, 0.916) | 0.003 | -53.2% |
| I | **Assault (48.9, 7.28)** | **Assault (39.4, 7.14)** | $\mathbf{1.07 \times 10^{-5}}$ | **-19.4%** |
| I | Battery (117, 21.7) | Battery (100, 15.15) | 0.004 | -14.5% |
| I | **Criminal Sexual Assault (4.09, 2.50)** | **Criminal Sexual Assault (1.80, 1.08)** | $\mathbf{7.76 \times 10^{-4}}$ | **-56.0%** |
| I | Homicide (1.15, 1.08) | Homicide (0.733, 0.704) | 0.155 | -36.3% |
| I | Sex Offense (2.91, 1.65) | Sex Offense (2.07, 1.49) | 0.067 | -28.9% |
| | **Baltimore** | | | |
| | **Total Crimes (103, 15.4)** | **Total Crimes (76.3, 12.4)** | $\mathbf{6.39 \times 10^{-9}}$ | **-25.9%** |
| | Gun Crimes (12.8, 6.70) | Gun Crimes (11.1, 6.21) | 0.359 | -13.3% |
| P | Arson (0.216, 0.441) | Arson (0.133, 0.352) | 0.493 | -38.4 |
| P | **Auto Theft (8.85, 2.68)** | **Auto Theft (6.20, 2.70)** | $\mathbf{6.01 \times 10^{-4}}$ | **-29.9%** |
| P | **Burglary (11.4, 4.07)** | **Burglary (8.07, 2.52)** | **0.003** | **-29.2%** |
| P | **Larceny (34.3, 8.83)** | **Larceny (22.3, 4.68)** | $\mathbf{1.24 \times 10^{-6}}$ | **-35.0%** |
| P | Robbery (12.1, 4.30) | Robbery (8.73, 4.64) | 0.007 | -27.9% |
| P | Shooting (1.49, 1.68) | Shooting (1.33, 1.35) | 0.735 | -10.7% |
| I | Assault (33.2, 7.84) | Assault (28.3, 6.74) | 0.024 | -14.8% |
| I | Homicide (0.761, 1.13) | Homicide (0.933, 0.799) | 0.575 | 22.6% |
| I | Rape (0.511, 0.773) | Rape (0.333, 0.488) | 0.391 | -34.8% |
| | **Baton Rouge** | | | |
| | **Total Crimes (113, 20.0)** | **Total Crimes (87.7, 15.8)** | $\mathbf{1.42 \times 10^{-5}}$ | **-22.4%** |
| P | Burglary (13.7, 5.75) | Burglary (12.1, 4.40) | 0.296 | -11.7% |
| P | Criminal Damage (9.03, 3.59) | Criminal Damage (10.0, 3.82) | 0.342 | 10.7% |
| P | Firearm (5.21, 3.82) | Firearm (4.53, 2.59) | 0.511 | -13.1% |
| P | Robbery (1.31, 1.23) | Robbery (1.27, 1.44) | 0.898 | -3.05% |
| P | Theft (29.3, 7.39) | Theft (24.7, 7.47) | 0.028 | -15.7% |
| S | Juvenile (1.13, 1.27) | Juvenile (0.867, 0.834) | 0.450 | -23.3% |
| S | **Narcotics (9.23, 5.97)** | **Narcotics (4.40, 2.72)** | **0.003** | **-52.3%** |
| S | Nuisance (1.78, 1.26) | Nuisance (1.33, 1.72) | 0.245 | -25.3% |
| S | **Other (23.4, 8.94)** | **Other (13.7, 6.01)** | $\mathbf{1.06 \times 10^{-4}}$ | **-41.5%** |
| S | Vice (0.575, 0.708) | Vice (0.600, 0.910) | 0.905 | 4.35% |

*(Continued)*

**Table 2.** (Continued)

| I | Assault (6.45, 3.03) | Assault (5.40, 1.84) | 0.198 | -16.3% |
|---|---|---|---|---|
| I | Battery (9.84, 4.38) | Battery (7.00, 1.96) | 0.016 | -28.9% |
| I | Homicide (1.20, 1.24) | Homicide (1.53, 0.916) | 0.324 | 27.5% |
| I | Sexual Assault (0.512, 0.675) | Sexual Assault (0.333, 0.488) | 0.330 | -35.0% |

were more stark significant differences between past years and 2020 (see S12-S14 Tables in S1 File). Total crimes during this time period declined compared to the past three years. There are a number of different crime types contributing to this decline, and few of the contributing crime types are interpersonal (see S7-S10 and S14 Figs in S1 File). The exceptions are significant declines in assaults and batteries between 2017 and 2019 compared to 2020 (S12 and S14 Tables in S1 File). These results show that the crime dynamics of 2020 are significantly different from those in past years and that the changes would not have been expected based on time-based observations prior to the pandemic outbreak.

We also compared the crime numbers in Chicago between different time periods. Comparing crime types before the stay at home order was put in place to the two weeks after implementation indicates that most of the crime types tested showed significant declines. They include total crimes, assaults, burglaries, criminal sexual assaults, criminal trespasses, interference with public officers, narcotics, other offenses, robberies, and thefts (Table 2, Fig 2, S14 Fig in S1 File). This shows that the immediate time period after stay at home orders were announced does correlate with decreased total crimes, but that the crimes contributing to this decline are related more to property crimes and statutory crimes than to interpersonal crimes (S7-S10 Figs in S1 File).

To determine whether this pattern holds in other cities, we also carried out time period comparisons for Baltimore, MD and Baton Rouge, LA (Table 2). Both cities showed similar results. In Baltimore, comparing the time period before the stay at home order was implemented and the two weeks after shows that total crimes, auto thefts, burglaries, and larceny showed significant differences (Fig 2, S14 Fig in S1 File). For Baton Rouge, total crimes, narcotics, and other crimes showed significant differences when comparing the two weeks after the stay at home order was put in place and the time period before the order was implemented (Fig 2, S14 Fig in S1 File).

In all of our tests across different years and different time periods, we find that the implementation of social distancing and quarantine protocols led to significant decreases in crime in the first two weeks. There were declines during the state-of-emergency time period (March 9—March 21), but they were not statistically significant (S7 Table in S1 File). Total crimes declined in Chicago after the stay at home order was put in place, but the crime types contributing to this decline are mostly property-based and statutory rather than interpersonal (Fig 2, S7-S10 and S14 Figs in S1 File). Similar patterns hold for both Baltimore, Maryland and Baton Rouge, Louisiana. Both cities showed declines in crimes after the introduction of stay at home orders. As in Chicago, the crime types contributing to this decline are more often property-based or statutory rather than individuals (S11 and S12 Figs in S1 File). These patterns may have only been present in the initial time period following implementation and may not persist over longer time periods.

The observed patterns could be the result of several different mechanisms. First, there may be differences in policing and reporting under stay at home orders. For interpersonal crimes, victims who are quarantined with their abusers may be less likely to speak up. Social distancing

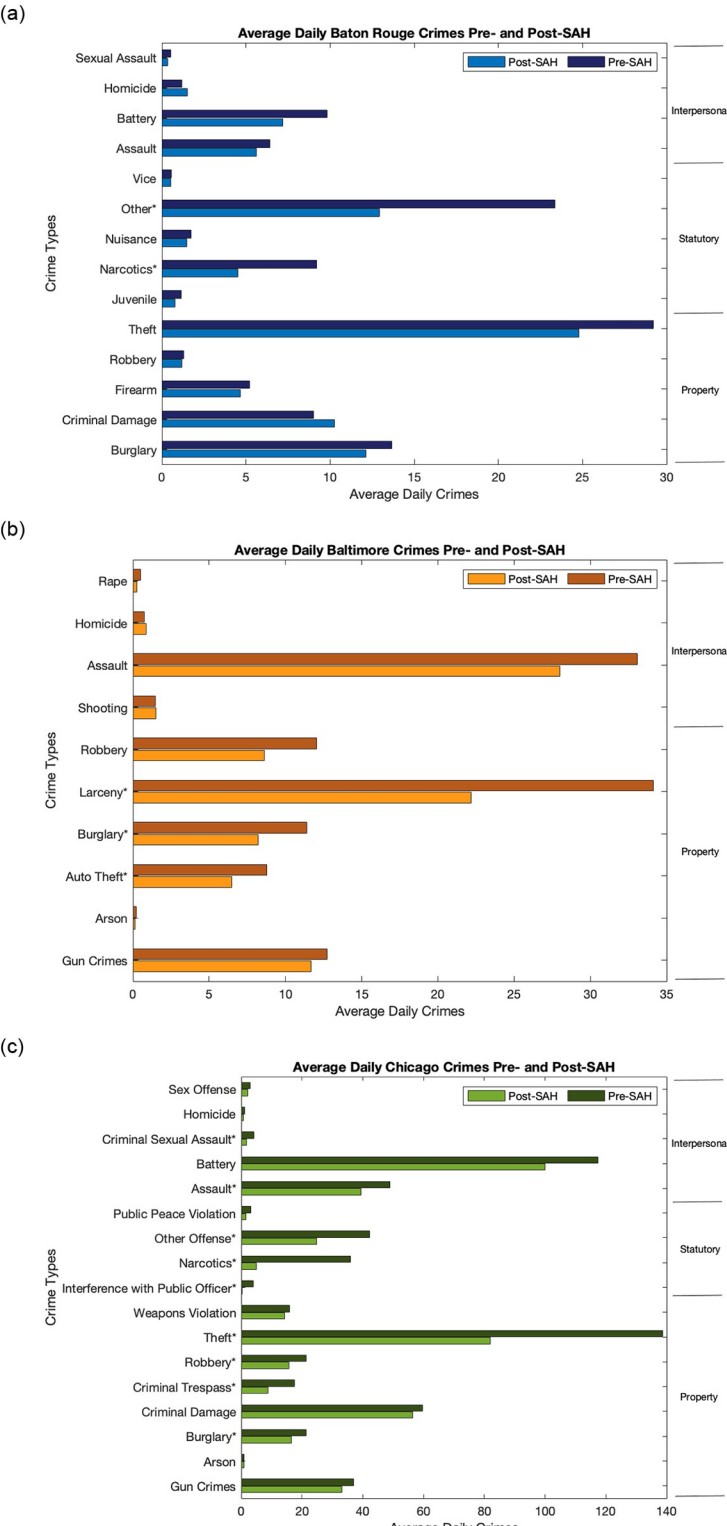

**Fig 2. All crime types across Chicago, Baltimore, and Baton Rouge split into pre- and post-stay-at-home order time periods.** We also split the crime types into the three crime categories: interpersonal, statutory, and property. The crime types that show significant differences are denoted with an asterisk.

may prevent law enforcement officers from responding to and reporting certain crime types. This pattern may also arise from individuals spending less time in public spaces and therefore participating in social interactions. The exact reasons contributing to this change in crime dynamics are difficult to determine, but it is clear that in the immediate time period following implementation of stay at home orders, there was a significant change in crime in Chicago, Baltimore, and Baton Rouge.

This study has a number of limitations. There is differential data availability across the three cities. Chicago has far more crime types publicly accessible for analysis, and there is no consensus on how the data are collected and reported nationally. We have also only analyzed the data for three cities. This pattern therefore may not exist across all regions, especially if adherence to stay at home orders differs between cities. The use of $t$-tests also limits the information we can obtain from these data. Finally, we have only observed the two weeks following the implementation of stay at home orders. The declines in crime may be limited to this time period and not subsist. With this analysis, we cannot pinpoint exactly when the decline began and when crime returned to previously expected levels. Recent reports show that violent crime has increased as cities are reopening [6].

These limitations leave open questions for future research. First, there is a need to explore the data presented here more in depth using various statistical tests, expanding the temporal window, or exploring confounding variables in all three cities. Additionally, analysis of patterns of crime data from additional cities is necessary to verify that the observed changes are national in scope. There is also a need to determine some of the mechanisms which have produced these declines and how adherence to stay at home orders impacts the crimes which occur. Overall, stay at home orders produced declines in crime over the initial time period and the crimes contributing to this decline were mainly property-based. The COVID-19 pandemic responses present a forced social experiment impacting many behavioral components and providing opportunities for novel explorations of the connection between behavioral constraints and crime. Given our results, the vigorous public policy debates regarding the impacts of potential interventions on violent crime, particularly gun crime, could benefit from further detailed analysis of imposed regulations arising from the pandemic.

## Supporting information

**S1 File. Complete supplementary text, figures, and tables.**
(PDF)

## Acknowledgments

The authors acknowledge the participants of the NIMBioS Investigative Workshop on the Mathematics of Gun Violence and two anonymous reviewers who provided feedback on a previous version of this manuscript.

## Author Contributions

**Conceptualization:** Shelby M. Scott, Louis J. Gross.

**Data curation:** Shelby M. Scott.

**Formal analysis:** Shelby M. Scott.

**Funding acquisition:** Louis J. Gross.

**Methodology:** Shelby M. Scott, Louis J. Gross.

**Resources:** Louis J. Gross.

**Supervision:** Louis J. Gross.

**Validation:** Shelby M. Scott.

**Visualization:** Shelby M. Scott.

**Writing – original draft:** Shelby M. Scott.

**Writing – review & editing:** Louis J. Gross.

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
