## [Decision Letter · Decision Letter 0]

11 Nov 2020

PONE-D-20-31968

COVID-19 and crime: Analysis of crime dynamics amidst social distancing protocols

PLOS ONE

Dear Dr. Scott,

Thank you for submitting your manuscript to PLOS ONE. Please see below for comments from the referees. They have some concerns about methodology, clear communication of results, and a few other issues. The referees have thus suggested various helpful methodological and stylistic improvements.

You'll see that the referees have both recommended "major revision" for your manuscript, and I agree with this assessment. I feel moved to say that I think the type of work you have done is extremely valuable and so I sincerely hope you will choose to undertake the revision.

We look forward to receiving your revised manuscript.

Cheers and best,

Chad

Chad M. Topaz

Academic Editor

PLOS ONE

2. Please include captions for your Supporting Information files at the end of your manuscript, and update any in-text citations to match accordingly. Please see our Supporting Information guidelines for more information: http://journals.plos.org/plosone/s/supporting-information

Reviewers' comments:

Reviewer's Responses to Questions

**Comments to the Author**

1. Is the manuscript technically sound, and do the data support the conclusions?

Reviewer #1: Yes

Reviewer #2: Partly

2. Has the statistical analysis been performed appropriately and rigorously? 

Reviewer #1: Yes

Reviewer #2: I Don't Know

3. Have the authors made all data underlying the findings in their manuscript fully available?

Reviewer #1: Yes

Reviewer #2: Yes

4. Is the manuscript presented in an intelligible fashion and written in standard English?

Reviewer #1: No

Reviewer #2: Yes

5. Review Comments to the Author

Reviewer #1: The authors aim to analyze the effect of COVID-19 on crime by studying a data set from Chicago, and comparing to similar data sets in Baltimore and Baton Rouge. They used t-tests and the Bonferroni correction to determine which crimes were significantly affected by stay-at-home orders due to COVID-19. The authors found that the greatest change in behavior of crime occurred in Chicago, where they compared crime types in early 2020 to those same crimes in earlier years (2019, 2018 and 2017). Then, they compared changes in crime between three time periods in 2020 across the three cities. The authors found that many crimes did indeed have a significant decrease after the stay-at-home orders were imposed in each city and that most crimes that were found to have significantly decreased were property crimes, not interpersonal crimes.

I have some questions and concerns with the organization of the paper:

- I don’t understand the choice to only analyze two weeks following the stay-at-home orders. I think the paper would be significantly strengthened by including more data after the stay-at-home orders. We are now well past the end date of those orders, so it makes sense to check and see if crime indeed went back up (or didn’t).

- I feel like the authors were very thorough in comparing changes in the 2020 data to previous years to make sure the effects were not due to seasonality for Chicago, but why not for the other two cities? Is it fair to only use the 2020 data for those two cities after performing such an in-depth analysis of Chicago to claim that the changes in their crimes were in fact due the stay-at-home order?

- Why scatter plots? They are super hard to read. For the t-test, aren’t you just taking total number before and after? Why not show just that? Or a moving average if you want to show time effects?

- What were the stay-at-home orders in the different cities? Did they all have similar stay-at-home orders Similar punishments for breaking the order? Any way to gauge how well people actually listened?

- The introduction seems to concentrate on the idea of interpersonal vs. property crimes, but there doesn’t seem to be much discussion or analysis of this in the text. Perhaps a visualization here would be nice where you color-code crimes that are considered personal and property and show in which city they increased, decreased or stayed the same.

-All of the tables in the main text and the SI are too hard to digest. I think summarizing the results in figures in the main paper and referencing those tables might be better.

In general I feel that this paper would be much easier to understand if the authors put some thought into creating figures that summarize the data and support their conclusions. Here are some suggestions:

- For Figure 1, plot the total crime in first three months of 2020 compared to 2019 and 2018 and compute the percent change.

- For Figure 2, color code the crimes as property or interpersonal and plot the change in number of crimes for each city that were statistically significant. I’d like to know from this graph, which crimes had similar behavior across the three cities and by how much did they change

- In the SI text, a comparison of the total number (or average number) of crimes between same time periods of 2019 compared to 2020 (summarizing Tables S1 - S7) would be good. You can even include this in the first figure summarizing Table 1 by breaking down the first three months to the three time periods that are analyzed in the SI text.

Again, I wouldn’t use scatter plots to visualize the data since the actual statistical analysis that’s being performed is simply on the total number of crimes before and after a date.

Smaller concerns:

- The introduction is lacking a bit and I would appreciate a bit more motivation for choosing these three particular crime sets, what they have in common, and what you found (be specific, by how much did crime overall seem to change during the stay-at-home order?).

- Write out less than or equal to rather than using the math symbol within paragraphs (e.g., lines 68, 123, 124)

- Additional and in line 133

- Say what sigma and mu are in Table 1 and 2 caption.

- A but more explanation on the Bonferonni correction and what you are using in the t-tests might be nice for readers that don't have a background in statistics

Reviewer #2: Please see the document attached above for my commentary on this paper. I forgot to mention some typos spread throughout the paper and things like "and and" that the authors should check on their manuscript.

6. PLOS authors have the option to publish the peer review history of their article (what does this mean?). If published, this will include your full peer review and any attached files.

Reviewer #1: No

Reviewer #2: No

---

## [Author Response · Author response to Decision Letter 0]

24 Dec 2020

Dear Dr. Topaz,

First, we would like to thank you and the two anonymous reviewers for the comments and for the

helpful feedback provided, which has enhanced the manuscript in many ways. We greatly appreciate the

effort that the reviewers made to constructively provide advice about the manuscript and believe the

resulting aper is much stronger due to their reviews. Below is our response to the comments provided on

our manuscript entitled, “COVID-19 and crime: Analysis of crime dynamics amidst social distancing

protocols.” Our responses follow each comment.

Reviewer #1 Comments

1) I don’t understand the choice to only analyze two weeks following the stay-at-home orders. I

think the paper would be significantly strengthened by including more data after the stay-athome

orders. We are now well past the end date of those orders, so it makes sense to check

and see if crime indeed went back up (or didn’t).

We chose to analyze the two weeks following the implementation of stay-at-home orders due

to the likelihood that stronger adherence to stay at home orders would occur in the initial

time period following the announcement (see figures S1-S3). From the data, all three metrics

showed the best adherence to social distancing protocols immediately following their

implementation. The crime data from 2020 (see figures S4-S6) also show that April was the

month with the lowest number of crimes. Therefore, determining whether there is a

correlation between social distancing protocols and crime will be most clear when tested in

this time period. We are also aware that the full time period should be tested now that more

data are available but wanted to highlight the initial impacts of these protocols. We have

added more explanation within the manuscript as to our decision to test these two weeks,

which is further supported by new figures within the supplemental information.

2) I feel like the authors were very thorough in comparing changes in the 2020 data to previous

years to make sure the effects were not due to seasonality for Chicago, but why not for the

other two cities? Is it fair to only use the 2020 data for those two cities after performing such

an in-depth analysis of Chicago to claim that the changes in their crimes were in fact due the

stay-at-home order?

Chicago is often mentioned as a major hub for gun crime in the United States, making it an

important study system for crime in general. Also, it was one of the major cities in the United

States which experienced an early COVID-19 outbreak. The initial purpose of this study was

to observe changes that occurred in Chicago following the implementation of stay-at-home

orders, but we then determined that it would be helpful to determine whether these patterns

held across other cities. To address this in the manuscript, we have altered our paper

objectives to state that Chicago is our main study area and that the comparisons with

Baltimore and Baton Rouge were to support our conclusions about Chicago. We have also

added line plots highlighting the temporal changes in Baltimore and Baton Rouge over the

last three years to allay concerns that differences in the stay-at-home time period were due to

seasonality. In the supplementary material, the line plots added also include monthly total

crime data for 2020 through October as a basis for potential future discussions of longer term

impacts of behavior on total crime.

3) Why scatter plots? They are super hard to read. For the t-test, aren’t you just taking total

number before and after? Why not show just that? Or a moving average if you want to show

time effects?

We chose to use scatter plots in order to highlight the variation present in the datasets and

to highlight the differences in pre- vs. post-stay-at-home order crime. We have improved the

clarity of the scatter plots and added in a 5-day moving average trend line and discussed the

calculation of this in the supplementary material. This moving average analysis did not add

any new conclusions but may improve clarity of the underlying trends for readers. In terms of

the t-tests, we have further clarified the objectives of the t-tests to compare the average

number of crimes per day pre- and post-stay-at-home orders (as well as for comparisons of

time periods from previous years over the same seasons).

4) What were the stay-at-home orders in the different cities? Did they all have similar stay-athome

orders Similar punishments for breaking the order? Any way to gauge how well people

actually listened?

The stay-at-home orders were generally the same across the three cities. They closed all

non-essential businesses, mandated wearing masks, and encouraged citizens to minimize

non-essential visits outside their homes. The punishments for stay-at-home orders included

fines or potential jail time in Maryland and Illinois, but Louisiana did not enforce fines.

Information regarding the mandates and adherence to stay-at-home orders have been added

to the manuscript. We note that in the three metrics for adherence (change in distance

traveled, non-essential visits, and encounter density) across all three cities, the lowest values

over the year (through November) occurred during the initial time period following

implementation of stay-at-home-orders (see figures S1-S3). We have added figures

highlighting adherence to the supplemental information.

5) The introduction seems to concentrate on the idea of interpersonal vs. property crimes, but

there doesn’t seem to be much discussion or analysis of this in the text. Perhaps a

visualization here would be nice where you color-code crimes that are considered personal

and property and show in which city they increased, decreased or stayed the same.

This is a great suggestion. The crime types for each city are now broken down into

interpersonal vs. property vs. statutory crimes using (1, 2, and 3) to improve understanding

within the manuscript. The specific crime types are defined in the supplementary material

(table S2). We created a bar chart of the different crime types pre- and post-stay-at-home

orders that is now in the supplementary material (see figures S7-S12) to show how crime

categories shifted over these time periods. We have also added a statement regarding the

lack of uniformity of crime data availability across the U.S., which constrains the ability to

match exact crime types between cities.

6) All of the tables in the main text and the SI are too hard to digest. I think summarizing the

results in figures in the main paper and referencing those tables might be better.

We have constructed bar charts showing the differences in the three crime categories

between time periods (see figures S7-S12). We have also added the percent and direction

change to the tables in order to clarify the importance of each crime type and category’s

contributions to the overall decline in total crimes across the three cities.

7) In general, I feel that this paper would be much easier to understand if the authors put some

thought into creating figures that summarize the data and support their conclusions. Here are

some suggestions:

a. For Figure 1, plot the total crime in first three months of 2020 compared to 2019 and

2018 and compute the percent change.

This is a great suggestion. The percent change has been included for each table (see

tables 1 and 2, S3-S19).

b. For Figure 2, color code the crimes as property or interpersonal and plot the change in

number of crimes for each city that were statistically significant. I’d like to know

from this graph, which crimes had similar behavior across the three cities and by how

much did they change.

We have created bar charts for changes pre- and post-stay-at-home-order for the three

crime categories in each of the three cities to allow comparisons across cities (see

figures S7-S12). Due to the lack of uniformity in publicly available data, comparisons

of exact crime types between cities are difficult (e.g., theft data in Chicago differs from

auto theft data in Baltimore), but we hope the inclusion of crime categories and

percent change within the table clarifies the patterns observed in the data.

8) In the SI text, a comparison of the total number (or average number) of crimes between same

time periods of 2019 compared to 2020 (summarizing Tables S1 - S7) would be good. You

can even include this in the first figure summarizing Table 1 by breaking down the first three

months to the three time periods that are analyzed in the SI text. Again, I wouldn’t use scatter

plots to visualize the data since the actual statistical analysis that’s being performed is simply

on the total number of crimes before and after a date.

The purpose of a t-test is to compare the means across the two time periods. The data show

the day-to-day variability in occurrence of each crime. Scatter plots allow this variability to

be demonstrated for readers to encourage appreciation of this variability in cases for which

there are significant differences seen across the two time periods. We have added line graphs

showing the monthly total crime dynamics across the three cities in each year 2017-2019 and

January-October of 2020 (see figures S4-S6). We have also shown in tables S3-S19 the

percent change to further clarify the dynamics between the time periods of interest.

Smaller concerns:

9) The introduction is lacking a bit and I would appreciate a bit more motivation for choosing

these three particular crimes sets, what they have in common, and what you found (be

specific, by how much did crime overall seem to change during the stay-at-home order?).

We have added more information in the introduction to clarify the motivation and

conclusions of this study for both reviewers.

10) Write out less than or equal to rather than using the math symbol within paragraphs (e.g.,

lines 68, 123, 124).

This has been updated.

11) Additional and in line 133

This change has been made.

12) Say what sigma and mu are in Table 1 and 2 captions.

Mu and sigma have been defined in the table captions.

13) A bit more explanation on the Bonferroni correction and what you are using in the t-tests

might be nice for readers that don't have a background in statistics.

Further explanation of t-tests and the Bonferroni correction have been added to the

manuscript.

Reviewer #2 Comments

1) For example, for Chicago they use 20 out of 32 types of crime for their statistics, with a

cutoff of each crime being at least 0.2 percent of total tally. For the other cities, no criteria

are presented. The crime lists don’t include anything else than the names of the crimes (we

don’t get to learn about numbers, locations, definitions, which ones are considered felonies

etc). In some case the nomenclature is bizarre - is “juvenile” a crime? How is “vice” defined?

How are “gun crimes” different than “shooting”? What is the crime of “criminal damage”?

For Baton Rouge and Baltimore, all crime types available have been included. Meanwhile,

the Chicago dataset has far more crime types available. Some of them only have one or two

occurrences throughout the entire year and no occurrences during the time periods we were

observing in this study. To determine the appropriate percentage of total crime cutoff, we

found the crime type in Baltimore or Baton Rouge with the lowest percentage of total crimes

and used this as the baseline to determine our cutoff for Chicago. We have highlighted this

explanation in the manuscript for clarity. Each dataset includes different information since

there is no consensus on how to collect, define, and present publicly available crime data.

We have added a statement about data quality to the manuscript and included definitions for

each crime type in the supplemental information (see table S2).

2) Similarly, the t-Test section is poorly presented. We are not told what alpha is, what is n,

how the Bonferroni (or is it Bonferonni? It is spelt in two different ways) correction is

supposed to affect the t-Test. They speak of an experiment within the context of the t-Test,

but it is not clear what experiment they are referring to. What is a “year in year” test? Later

they call it a “pair-wise t-test appropriately corrected for replication” without any context of

what the replication is.

The t-test is a comparison of means from two different populations (in this case, different

time periods), in order to determine whether the data come from the same distribution. Alpha

is the level of significance desired, n is the total size of our sample, and the Bonferroni

correction keeps random results from showing significance due to the number of t-tests we

are using. When a researcher completes a large volume of any sort of statistical test, there is

a random chance some of them will come up as significant due to repetition. The Bonferroni

correction helps to tighten the restrictions on what is significantly different in order to

prevent these errors due to replication. We have added further descriptions and definitions to

the manuscript in order to clarify the statistical methods used and the comparisons

undertaken. We have also corrected the spelling of Bonferroni within the manuscript.

3) Finally, in the results section they speak of “victim-based crime data from Baltimore”, what

exactly does this mean? Is it specific to Baltimore? Is crime from Chicago not victim-based?

The crime data available from Baltimore is narrowed to only crimes in which there are

victims. From the dataset, victim-based crime is defined as crimes in which someone is

victimized (either personally or in terms of property). Meanwhile, Chicago and Baton Rouge

have crime datasets that are comprised of more crime types. We have added clarifications

into the materials and methods about what each of these crime datasets include. We have

also divided the crime types into three different crime categories (property, statutory, and

interpersonal). These crime categories are highlighted in the tables (1 and 2), as well as

being used to create bar graphs that show shifts between pre- and post-stay-at-home order

time periods (see figures S7-S12). The supplemental information now also includes

descriptions of each crime type used from the datasets (see table S2).

4) They also discuss Baton Rouge as a “city with different demographics” what does this mean?

Is it in terms of race? age? wealth? Up until this point (page 5) there has been no discussion

on demographics of the different cities. Indeed, we are not even told what the population

numbers are.

Baton Rouge, Chicago, and Baltimore are all cities regularly studied for their crime

dynamics. Chicago is the largest city, followed by Baltimore and then Baton Rouge. Each of

them has differing poverty levels, but all fall above the national average (10.5%). Chicago is

the densest, followed by Baltimore and then Baton Rouge. Therefore, these cities have

similarities, but also differences that make them interesting comparisons. We have added

more information about the cities themselves into the descriptions of the materials and

methods to address this. We have also added a table of demographic information to the

supplemental material for further clarification (see table S1).

5) In Table 1 \\sigma and \\mu are not defined. Nor is the p-value.

All of these terms have been defined in the manuscript.

6) In the Conclusions it is not clear what their message is. Consider these two statements made

by the authors:

1) Chicago's total crimes during the first three months of the year (including the time period

following the stay-at-home order) declined in 2020 compared to 2017, 2018, and 2019.

2) Before COVID protocols were put in place, there were no significant changes in total

crimes compared to 2018 and 2019.

Since COVID-19 became a problem in March 2020, and all the protocols came after March

2020, what are we to believe? Did crime within January 2020-March 2020 decline (as the

first sentence would imply!) or not (as the second sentence would imply)?

Then they add nuances that confuse the reader even more:

3) There were some changes in particular crime types that may have been due to social

factors, policing protocols, or other mechanisms. There were also significant changes in total

crimes between 2017 and 2020, with all but weapons violations showing declines.

So, did all these crimes except for weapons violations decrease? Also, what does it mean that

there were changes in “crime types?” Did the definition change? This is unclear. They also

discuss interpersonal crimes, but do not define an interpersonal crime. Is robbery an

interpersonal crime? Unclear.

The major conclusion of this paper is that there were significant changes that occurred in the

two weeks after the stay-at-home orders were put in place across all three cities and that

more work needs to be done to analyze the impacts that changes in social dynamics have on

crime. This conclusion was supported by first making sure that 2020 was not showing

unexpected crime dynamics in Chicago before the pandemic began. We then checked to be

sure that there were not seasonal dynamics in place by comparing time periods of past years

to the crime dynamics in 2020 (see figures S4, S13). Once we were certain that, generally,

there are no existing temporal patterns at play, we then test to see whether the stay-at-home

orders had a significant impact on the crime dynamics in Chicago and then, for comparison,

two other cities. We have clarified the conclusions of the manuscript and added more figures

to the supplementary material for ease of reader understanding. Specifically, we have added

descriptions of crime types and data quality, divided the crime types into crime categories,

and carried these definitions throughout the manuscript.

Sincerely,

Shelby M. Scott

National Defense Science and Engineering Graduate Fellow

Department of Ecology and Evolutionary Biology

Louis J. Gross

Chancellor’s Professor and Alvin and Sally Beaman Distinguished Professor of Ecology and Evolutionary

Biology and Mathematics

Director, National Institute for Mathematical and Biological Synthesis

Director, The Institute for Environmental Modeling, University of Tennessee

Past-President, The Society for Mathematical Biology

---

## [Decision Letter · Decision Letter 1]

29 Jan 2021

PONE-D-20-31968R1

COVID-19 and crime: Analysis of crime dynamics amidst social distancing protocols

PLOS ONE

Dear Dr. Scott,

Thank you for submitting your manuscript to PLOS ONE. We appreciate the substantial improvements to the manuscript and one reviewer has signed off on it. The second reviewer has some additional constructive suggestions that we ask you to address. These suggestions are largely centered around bolstering the clarity of the argument you are making.

We look forward to receiving your revised manuscript.

Kind regards,

Chad M. Topaz

Academic Editor

PLOS ONE

Reviewers' comments:

Reviewer's Responses to Questions

6. Review Comments to the Author

Reviewer #1: The manuscript is much improved. The main takeaway of the article is much clearer. However, it’s still not clear to me that the figures and tables are the best way to support the hypothesis of the paper. It seems that the main point of the manuscript is that not all crimes changed in the same way after the stay-at-home (SAH) order; specifically, it seems that property and statutory crimes experienced significant decreases in number during the two weeks post SAH order, but interpersonal crimes did not.

Major points:

- The introduction should clearly state the main conclusion of the paper. It currently states “changes in crime dynamics in all three cities .. are not uniform .. there are differences” What are those differences? Being clear about which city had significant changes in which type of crime and by how much up front will make the rest of the paper easier to digest.

Some ways to better support the hypothesis in the text:

- Label the crime types using P, S, and I rather than (1), (2), and (3) so it is easier for the reader to understand

- Rearrange Table 1 and Table 2 to group crimes of a similar type rather than alphabetically. This way, the reader can easily see that the greatest change in crime was in P and S types, not I. Similarly for the list of crimes in each city in the Data section

- Re-think Figure 2. The purpose of this figure is unclear to me. It is referenced in the text (e.g., in line 249) as supporting the hypothesis that P and S type crimes significantly decreased during SAH orders. This figure, however, doesn’t clearly show that. Instead, it seems to show only two crimes that significantly decreased during this time period for each city. It doesn’t even state which category those crimes are in. I wonder if it would be more useful to show ALL crimes that had significant decreased in each city and again order them by crime type (or color code by crime type, not city) to see that they are mostly type S or P.

Figure 2a,b is not referenced in the text.

- Along the same lines as the point above, Fig S10-S12 does attempt to demonstrate this point (and is referenced in line 249), but is lacking. I would include all crime types for each city (also label them by type instead of number, or remind the reader in the caption what the number represents) and clearly label which exhibit significant decreases (this is typically done with an asterisk above the two bars). It might also be clearer to color code by crime type, not city, since the main point is that all three cities demonstrated the greatest change in similar crime types.

Other points:

- The t-test indicates that there was a significant change in the number of crimes before and after the SAH order was implemented, but the scatter plots show a decrease in crime count prior to this date. This point should be addressed in limitations. The type of analysis that was performed cannot determine the date at which the crime changed. For example, if you chose to perform the same test before and after 3/1 instead, it looks from the scatter data that there would still be a significant decrease in total crimes in Chicago (Fig 1a).

- It’s unclear what is considered as the time period before SAH (e.g., in caption of Table 2 there are dates for the SAH and the two weeks after. What are the dates for before? Jan 1 - 3/21?). This should be clearly stated.

- The SI text needs a bit more information to be readable. It should include more text in the caption of the tables and figures, or should just be put into the main manuscript. For example, Fig S4 — cited in line 190. It would be useful to include a sentence after the title of the figure to describe what the reader should take away from that figure.

Minor errors:

- There are inconsistencies in how the manuscript refers to the SI figs (e.g., line 233 vs. line 236) and Figure vs. Fig (for example, see Fig in line 249 and Figure in line 244).

- The font size in every table is way too small

- Extra 5 in line 27 and 14.

---

## [Author Response · Author response to Decision Letter 1]

9 Mar 2021

To whom it may concern,

First, we would like to thank the anonymous reviewer for their service and for the helpful feedback provided on our revision, which has enhanced the manuscript in many ways. Below is our response to the comments provided on our revision of the manuscript entitled, “COVID-19 and crime: Analysis of crime dynamics amidst social distancing protocols.” Our responses follow each comment in italics. Actions that have been taken or sections that have been added are colored in green.

Major points

1. The introduction should clearly state the main conclusion of the paper. It currently states “changes in crime dynamics in all three cities .. are not uniform .. there are differences” What are those differences? Being clear about which city had significant changes in which type of crime and by how much up front will make the rest of the paper easier to digest.

This is helpful feedback. We have explicitly listed the percent and direction change for each city’s overall crime, each of the crime types that significantly changed, and their percent and direction of change. The rest of the paper then outlines the support for these claims.

2. Some ways to better support the hypothesis in the text:

a. Label the crime types using P, S, and I rather than (1), (2), and (3) so it is easier for the reader to understand

We have added letter rather than number designations for crime categories throughout the manuscript.

b. Rearrange Table 1 and Table 2 to group crimes of a similar type rather than alphabetically. This way, the reader can easily see that the greatest change in crime was in P and S types, not I. Similarly, for the list of crimes in each city in the Data section.

We have rearranged the tables and lists within the manuscript to reflect this change.

c. Re-think Figure 2. The purpose of this figure is unclear to me. It is referenced in the text (e.g., in line 249) as supporting the hypothesis that P and S type crimes significantly decreased during SAH orders. This figure, however, doesn’t clearly show that. Instead, it seems to show only two crimes that significantly decreased during this time period for each city. It doesn’t even state which category those crimes are in. I wonder if it would be more useful to show ALL crimes that had significant decreased in each city and again order them by crime type (or color code by crime type, not city) to see that they are mostly type S or P.

Figure 2a, b is not referenced in the text.

Thank you for this incredibly helpful feedback. We have created a new Figure 2 that shows all of the crime types for the three cities, grouped by crime category, comparing average daily crime pre- and post-stay-at-home order implementation. The significant crime types are denoted with an asterisk next to the label. The past figure has been moved into the supplemental information (Fig S14) and now shows the time series with moving average trendline for all of the significant crime types in the three cities.

d. Along the same lines as the point above, Fig S10-S12 does attempt to demonstrate this point (and is referenced in line 249) but is lacking. I would include all crime types for each city (also label them by type instead of number or remind the reader in the caption what the number represents) and clearly label which exhibit significant decreases (this is typically done with an asterisk above the two bars). It might also be clearer to color code by crime type, not city, since the main point is that all three cities demonstrated the greatest change in similar crime types.

We have added a new Figure 2 that includes all the available crime types, grouped by crime category and have indicated which show significant different using an asterisk. We have maintained the color schemes based on cities for continuity between the main text and supplemental information but have split the crime types into crime categories throughout the main text for ease of understanding.

Other points

1. The t-test indicates that there was a significant change in the number of crimes before and after the SAH order was implemented, but the scatter plots show a decrease in crime count prior to this date. This point should be addressed in limitations. The type of analysis that was performed cannot determine the date at which the crime changed. For example, if you chose to perform the same test before and after 3/1 instead, it looks from the scatter data that there would still be a significant decrease in total crimes in Chicago (Fig 1a).

This is a great point, and we agree that there was a visual decline prior to implementation of the stay-at-home order. In the supplementary information (Table S7), we show that total crimes did decline in the state-of-emergency time period between the pre-COVID and stay-at-home order windows, but that this decline was not significant. We have added this information to the manuscript and also added to our limitations that we cannot know the exact date of the change in dynamics.

2. It’s unclear what is considered as the time period before SAH (e.g., in caption of Table 2 there are dates for the SAH and the two weeks after. What are the dates for before? Jan 1 - 3/21?). This should be clearly stated.

The time period before spans from the beginning of the year 1/1/20 until before the stay-at-home order was implemented on 3/21/20. We have clarified this in the methods section of the manuscript.

3. The SI text needs a bit more information to be readable. It should include more text in the caption of the tables and figures or should just be put into the main manuscript. For example, Fig S4 — cited in line 190. It would be useful to include a sentence after the title of the figure to describe what the reader should take away from that figure.

We have added more text to the supplementary information and also have updated the captions to make the figures capable of being understood without having to refer to the supplement text. Because of journal limitations, we unfortunately cannot add more figures or tables into the main text of the manuscript.

Minor errors

1. There are inconsistencies in how the manuscript refers to the SI figs (e.g., line 233 vs. line 236) and Figure vs. Fig (for example, see Fig in line 249 and Figure in line 244).

Thank you for catching these discrepancies. They have been updated.

2. The font size in every table is way too small

We have increased the font size for the tables in the main manuscript before forcing a split onto multiple pages.

3. Extra 5 in line 27 and 14.

This has been rectified.

---

## [Decision Letter · Decision Letter 2]

18 Mar 2021

COVID-19 and crime: Analysis of crime dynamics amidst social distancing protocols

PONE-D-20-31968R2

Dear Dr. Scott,

We’re pleased to inform you that your manuscript has been judged scientifically suitable for publication and will be formally accepted for publication once it meets all outstanding technical requirements.

Also, you'll see a few typo corrections suggested by one reviewer.

Kind regards,

Chad M. Topaz

Academic Editor

PLOS ONE

Additional Editor Comments (optional):

Reviewers' comments:

Reviewer's Responses to Questions

**Comments to the Author**

1. If the authors have adequately addressed your comments raised in a previous round of review and you feel that this manuscript is now acceptable for publication, you may indicate that here to bypass the “Comments to the Author” section, enter your conflict of interest statement in the “Confidential to Editor” section, and submit your "Accept" recommendation.

Reviewer #1: All comments have been addressed

2. Is the manuscript technically sound, and do the data support the conclusions?

Reviewer #1: (No Response)

3. Has the statistical analysis been performed appropriately and rigorously? 

Reviewer #1: (No Response)

4. Have the authors made all data underlying the findings in their manuscript fully available?

Reviewer #1: (No Response)

5. Is the manuscript presented in an intelligible fashion and written in standard English?

Reviewer #1: (No Response)

6. Review Comments to the Author

Reviewer #1: I’m happy with the edits the authors have made. Here are just a few small editorial comments:

Line 27: should read “main focus is Chicago”

Line 186 should read S3-S5 Figs

Line 199: should say S4 Fig

Line 216: $t$-tests

7. PLOS authors have the option to publish the peer review history of their article (what does this mean?). If published, this will include your full peer review and any attached files.

Reviewer #1: No

---

## [Editor Report · Acceptance letter]

22 Mar 2021

PONE-D-20-31968R2 

COVID-19 and crime: Analysis of crime dynamics amidst social distancing protocols 

Dear Dr. Scott:

I'm pleased to inform you that your manuscript has been deemed suitable for publication in PLOS ONE. Congratulations! Your manuscript is now with our production department. 

Kind regards, 

on behalf of

Dr. Chad M. Topaz 

Academic Editor

PLOS ONE